# Physiological and Proteomic Analyses of *mtn1* Mutant Reveal Key Players in Centipedegrass Tiller Development

**DOI:** 10.3390/plants13071028

**Published:** 2024-04-04

**Authors:** Chenming Xie, Rongrong Chen, Qixue Sun, Dongli Hao, Junqin Zong, Hailin Guo, Jianxiu Liu, Ling Li

**Affiliations:** 1Jiangsu Key Laboratory for the Research and Utilization of Plant Resource, Institute of Botany, Jiangsu Province and Chinese Academy of Sciences (Nanjing Botanical Garden Mem. Sun Yat-Sen), Nanjing 210014, China; 15163726024@163.com (C.X.); 670702crr@163.com (R.C.); hndongli@163.com (D.H.); zongjq1980@163.com (J.Z.); ghlnmg@sina.com (H.G.); turfunit@aliyun.com (J.L.); 2College of Animal Science and Technology, Yangzhou University, Yangzhou 225009, China; mx120210904@stu.yzu.edu.cn

**Keywords:** tillering, mutant, centipedegrass, proteomic, plant hormones, sugars

## Abstract

Tillering directly determines the seed production and propagation capacity of clonal plants. However, the molecular mechanisms involved in the tiller development of clonal plants are still not fully understood. In this study, we conducted a proteome comparison between the tiller buds and stem node of a multiple-tiller mutant *mtn1* (*more tillering number 1*) and a wild type of centipedegrass. The results showed significant increases of 29.03% and 27.89% in the first and secondary tiller numbers, respectively, in the *mtn1* mutant compared to the wild type. The photosynthetic rate increased by 31.44%, while the starch, soluble sugar, and sucrose contents in the tiller buds and stem node showed increases of 13.79%, 39.10%, 97.64%, 37.97%, 55.64%, and 7.68%, respectively, compared to the wild type. Two groups comprising 438 and 589 protein species, respectively, were differentially accumulated in the tiller buds and stem node in the mtn1 mutant. Consistent with the physiological characteristics, sucrose and starch metabolism as well as plant hormone signaling were found to be enriched with differentially abundant proteins (DAPs) in the *mtn1* mutant. These results revealed that sugars and plant hormones may play important regulatory roles in the tiller development in centipedegrass. These results expanded our understanding of tiller development in clonal plants.

## 1. Introduction

Centipedegrass (*Eremochloa ophiuroides* (Munro) Hack.) is an important perennial warm-season ecological grass [1]. Centipedegrass exhibits favorable traits such as high tolerance to infertile conditions, soil consolidation, heavy metal adsorption, and ornamental value, making it widely utilized in soil conservation and environmental remediation [2]. Centipedegrass, a perennial turfgrass, spreads through stolons and seeds, both of which depend on the continuous tillering of stolon nodes [1,2]. Stolons are often observed in several perennial and clone plants, due to their important roles in ecological invasion [3,4,5]. Currently, the molecular regulatory mechanism underlying tiller development in centipedegrass is still unknown.

Tillering, which encompasses the initiation and outgrowth of tiller buds, is closely associated with seed yield and propagation capacity in clonal plants [1,6,7]. The development of tillering is influenced by several factors, such as genetic traits, hormones, nutrients, and water [5]. Plant hormones, especially auxin, play a vital role in tiller development [8]. Auxin, synthesized in the shoot apex, and transported by polar carriers, indirectly suppresses the outgrowth of axillary bud [9,10]. Overexpression of *EoABCB11* (*ATP*-*binding cassette B 11*) and the inhibition of *PvPIN1* (*PIN*-*FORMED 1*) gene expression increase tiller number/branch in Arabidopsis and switchgrass, respectively [11,12]. The rice double mutant, *lax1*/*lax2,* exhibits reduced tillering [13]. Our previous research demonstrated a decrease in auxin content in the *mtn1* mutant in centipedegrass, which consequently inhibited tillering, and the exogenous application of auxin further suppressed tiller development [11]. Gibberellins (GAs) play a role in regulating tillering/branch development, but their effects vary across species, such as centipedegrass, tall fescue, poplar and sweet cherry [11,14,15,16]. Previous studies reported that GAs inhibit tillering in annual herbaceous plants such as tall fescue [16], orchardgrass [17], and cultivated rice [18]. Exogenous application of GA3 inhibits the outgrowth of axillay buds by increasing the expression level of *FaGA3ox2* (*Gibberellin 3-β dioxygenase 2*) and *FaYUCCA9* (*Flavin monooxygenases 9*) in tall fescue [16]. The expression level of *DgCPS* and *DgGA2ox* (*Gibberellin 2-oxidases*) were down-regulated in orchardgrass cultivar with high tillering [17]. The GA receptor *GA*–*INSENSITIVE DWARF 1* (*GID1*) interacts with the DELLA protein, which functions as an inhibitor of GA [19]. Overexpression of *OsGID1* results in reduced tiller numbers in rice [20]. In contrast, GAs promote branch growth in perennial woody plants such as poplar [14] and sweet cherry [15]. Brassinosteroids (BRs), which are plant-specific steroid hormones, play a positively regulatory role in tillering by promoting bud outgrowth [21,22,23]. The BR-deficient mutants, *d11* (*dwarf11*) and *dlt*-*2* (*dwarf and low tillering 2*), displayed reduced tiller numbers compared to their wild type, while overexpression of *OsBZR1* increased tiller numbers in rice [24].

Sugars have been reported to play a signaling role in tillering across various species [25,26]. Sucrose, the primary product of photosynthesis and the main transport substance in plants, can be converted into hexose for tiller development [25,27]. Upon decapitation, pea buds exhibit a rapid accumulation of the low-abundant metabolite trehalose 6-phosphate (Tre6P), which promotes shoot branching in Arabidopsis [28,29]. Disruption of the sugar transporter *OsSTP15* enhances tiller numbers in rice [30]. Recently, studies have revealed that sucrose can antagonize the effects of auxin by inhibiting strigolactone perception, leading to the promotion of tillering [31]. Moreover, the growth-promoting effects of cytokinins may be limited when sugars are readily available [26].

The *mtn1* mutant was generated by our research team using cold plasma mutagenesis technology to induce somatic mutation in the Yuxi callus [11]. The *EoMTN1* gene encodes ABCB11, which belongs to the ATP-binding cassette protein subfamily B (ABCB) transporter family [11]. The study aimed to investigate the molecular regulatory mechanism of tiller development in centipedegrass by evaluating comparative phenotypic, physiological, and proteomic changes between the *mtn1* mutant and wild type. Specifically, the objectives were (1) to characterize physiological traits, such as photosynthetic activity, starch and soluble sugar contents, in the *mtn1* mutant and wild type; (2) to identify differentially accumulated proteins (DAPs) that contribute to the high tiller numbers in the *mtn1* mutant; and (3) to analyze the mechanisms underlying tiller development in centipedegrass by integrating physiological and proteomic changes.

## 2. Results

### 2.1. Phenotype of mtn1 Mutant

The tiller numbers of the *mtn1* mutant and its wild type (WT) were quantified and compared using statistical analysis. Compared to the WT, the tiller number of the *mtn1* mutant showed a significant increase, with a 29.03% higher count in the first tillering and a 27.89% higher count in the secondary tillering (Figure 1). These findings collectively demonstrate significant differences in tillering morphology between the *mtn1* mutant and the WT.

### 2.2. Physiological Characteristics

The results showed that the *mtn1* mutant accumulated higher levels of chlorophyll a and chlorophyll b contents compared to the WT (Figure 2a,b). The levels increased by 29.09% and 33.68%, respectively. The net photosynthetic rate of the *mtn1* mutant increased by 31.44% compared to the WT, consistent with the chlorophyll contents (Figure 2c). Additionally, the tiller bud and stem node of the *mtn1* mutant accumulated higher levels of starch, soluble sugar, and sucrose compared to the WT (Figure 2d–f). Starch levels increased by 13.79% and 39.10% in the tiller bud and stem node, respectively. Soluble sugar levels increased by 97.64% and 37.97% in the tiller bud and stem node, respectively. Sucrose levels increased by 55.64% in the tiller bud and 7.68% in the stem node.

### 2.3. Protein Data Quality

A label-free quantitation (LFQ) proteomic analysis was conducted to identify the DAPs between the mtn1 mutant and wild type of centipedegrass. Statistical and quality analyses were conducted on the mass spectrometry results. Through database searching, a total of 5166 proteins and 40,178 peptides were identified from 322,248 MS/MS spectra (Figure 3a). Among these proteins, 4833 were annotated for GO functions, while 3307 were annotated for KEGG functions (Figure 3b). Figure 3c demonstrates that out of the total 5,166 identified proteins, a significant portion was located in the cytoplasm (31.90%), chloroplasts (24.35%), and mitochondria (9.85%). Principal component analysis (PCA) revealed a clear separation between the *mtn1* mutant and WT, suggesting significant differences in proteins found in the tiller bud and stem node between the *mtn1* mutant and the WT (Figure 3d).

### 2.4. Differential Expression Proteins Analysis

In the tiller buds of the *mtn1* mutant, a total of 438 DAPs were identified, including 260 up-regulated proteins and 178 down-regulated proteins, when compared to WT (Figure 4a). In the stem node of the *mtn1* mutant, 589 DAPs were identified, including 349 up-regulated proteins and 240 down-regulated proteins, compared to the WT (Figure 4a). In the tiller buds, 194 proteins were specifically up-regulated and 103 proteins were specifically down-regulated, whereas in the stem node, 272 proteins were specifically up-regulated and 178 were specifically down-regulated (Figure 4b).

### 2.5. Differentially Regulated Biochemical Pathways of mtn1 Mutant and WT

GO and KEGG annotations were conducted to obtain potential functional information of the DAPs. GO annotation revealed that 398 DAPs were involved in 57 biological processes in the tiller bud, while 535 DAPs were involved in 140 biological processes in the stem node (Appendix A). In the tiller bud, 44 DAPs were enriched in cellular components associated with anchoring junctions, plasmodesma, and cell junctions. Regarding molecular functions, 33 DAPs were found to be involved in self-binding, unfolded protein binding, and chitinase activity. In terms of biological processes, 63 DAPs were enriched in protein complex oligomerization, stress response, and cell wall organization (Figure 5a, Appendix A). In the stem node, 121 DAPs were found to be enriched in cellular components associated with nucleus, nuclear protein complexes, and the cell cortex. Additionally, 66 DAPs were enriched in molecular functions related to protein self-association, protein kinase activity, and serine/threonine kinase activity. In terms of biological processes, 32 DAPs were observed to be related to protein complex oligomerization, stress response, and mRNA decay metabolism (Figure 5b, Appendix A).

The results of KEGG pathway enrichment analysis results revealed that the DAPs in the tiller buds of the *mtn1* mutant were primarily involved in endoplasmic reticulum protein processing, ribosome, glycine, serine, and threonine metabolism, oxidative phosphorylation, and the MAPK signaling pathway (Figure 6a). The DAPs in the stem node of the *mtn1* mutant were mainly associated with RNA polymerase, MAPK signaling pathway, glycerophospholipid metabolism, and photosynthetic carbon fixation (Figure 6b). The tiller buds and stem nodes exhibited a significant enrichment of proteins involved in metabolic pathways, including plant hormone signal transduction and starch and sucrose metabolism. This suggests that these pathways may play important roles in tiller development in centipedegrass.

### 2.6. DEPs Involved in Plant Hormone Signal Transduction

Seven DAPs (four up-regulated and three down-regulated) were found to be involved in plant hormone signal transduction pathways, including auxin, GAs, and BRs. Plant hormone signal transduction in the stem nodes involved six DAPs (two up-regulated and four down-regulated) (Appendix A, Figure 7). In the tiller buds, the auxin polar transport carrier BIG protein was up-regulated, while the IAA-amino acid hydrolase was down-regulated. Furthermore, three DAPs, namely ent-kaurene oxidase (KO), gibberellin receptor GID1L2, and geranylgeranyl diphosphate synthase (GGPS), were found to be involved in GA metabolism and signal transduction. Moreover, the typhasterol/6-deoxotyphasterol 2alpha-hydroxylase (CYP92A6) involved in BR synthesis was found to be up-regulated, while the BR-signaling kinase is down-regulated. In the stem node, two DAPs, namely AUX1/LAX and indole-3-acetaldehyde oxidase (AAO), were involved in auxin signal transduction. Additionally, two DAPs, the up-regulated gibberellin receptor GID1L2 and down-regulated geranylgeranyl diphosphate synthase (GGGPS), were identified to be involved in GA signal transduction. Finally, 2 DAPs, namely xyloglucosyl transferase 4 (TCH4) (up-regulated) and BR-signaling kinase (down-regulated), were found to be involved in BR signal transduction.

### 2.7. DAPs Involved in Starch and Sucrose Metabolism

DAPs involved in starch and sucrose metabolism were detected in both the tiller buds and stem nodes of the *mtn1* mutant. We identified a total of four DAPs in the tiller buds and stem nodes, with two being up-regulated and two being down-regulated. Additionally, six up-regulated DAPs were found in the tiller buds, while the stem nodes exhibited six up-regulated DAPs (Appendix A and Figure 8). Tiller buds showed up-regulation of β-glucosidase (GLU) and 1,4-α-glucan-branching enzyme (GBE), while down-regulation was observed for the glycosyl hydrolase family 9 (GH9) and β-amylase (βAMY). Conversely, the stem nodes exhibited up-regulation of βAMY, trehalose-6-phosphate synthase (TPS), endo-1,4-β-D-glucanohydrolase (EG), GBE, and GLU.

### 2.8. qRT-PCR

In order to analyze the proteomic data, a total of eight protein-coding genes were randomly selected. The primer sequences for both the internal reference gene and the selected gene can be found in Appendix A. The results demonstrate that the gene expression levels detected by qRT-PCR align with the protein expression levels in the proteome (Figure 9).

### 2.9. Summary Analysis of the Physiological Parameters and Proteomes

Plant hormones and sugars played crucial roles in regulating tiller development in centipedegrass, as indicated by the findings of the physiological and proteome analyses (Figure 10). Auxin and GAs negatively regulate tiller development, whereas BRs positively regulate tiller development. Hormone biosynthesis as well as transport and signal transduction are crucial for regulating cell proliferation, division, and elongation during tiller development. The increase in soluble sugar content not only provides additional energy, but also interacts with plant hormones to regulate tiller development in the *mtn1* mutant. The interaction between plant hormones and sugars is responsible for the observed increase in tillering in the *mtn1* mutant.

## 3. Discussion

The tiller number directly affects the seed yield and propagation capacity of clone plants [6,7]. Centipedegrass is a native grass species with considerable potential for ecological restoration, but its molecular mechanism affecting tiller development remains unknown [11]. This study conducted quantitative physiological and proteomic analyses of the tiller buds and stem nodes in *mtn1* mutant WT. The results showed an increase in chlorophyll content, net photosynthetic rate in leaves, and starch, soluble sugar, and sucrose contents in the tiller bud and stem node of the *mtn1* mutant (Figure 2). The IAA and GA contents in the tiller bud of the *mtn1* mutant were reduced [11]. A total of 438 and 589 DAPs were identified in the tiller buds and stem nodes, respectively (Figure 4a). The enriched proteins in the *mtn1* mutant were found to be involved in plant hormone signal transduction pathways and starch and sucrose metabolic pathways. Plant hormones, including auxin, GAs, and BRs, along with sugars, play a crucial role in regulating tiller development in plants. Therefore, we propose that these factors are significant in regulating tiller development in centipedegrass [8,26]. These results provide insights into tiller development in clonal plants.

Plant hormones play a crucial role in the regulation of tiller development [6]. This study found a significant enrichment of auxin, GA, and BR metabolism with DAPs in the tiller bud and stem node of the *mtn1* mutant (Figure 7, Appendix A). BIG exhibited accumulation in the *mtn1* mutant, while IAA–amino acid hydrolase (ILR1), indole-3-acetaldehyde oxidase (AAO), and AUX1/LAX were down-regulated in the *mtn1* mutant (Appendix A). These results were consistent with previous studies conducted on upland cotton, wheat, and matsudana, which also demonstrated a preferential expression of auxin synthesis enzymes and transporters in tillers/branches [32,33,34]. ILR1 converts IAA–amino acid conjugates (IAA–Asp and IAA–Glu) into auxin; AAO catalyzes the conversion of indole-3-acetaldehyde into auxin, while BIG and AUX1/LAX proteins serve as auxin efflux carriers and influx carriers, respectively [35,36,37,38]. The simultaneous high protein abundance of these auxin synthesis and transport proteins led to lower auxin levels in the *mtn1* mutant [11]. It is hypothesized that the *mtn1* mutant maintains lower IAA levels in the tiller buds and stem nodes due to reduced activity of ILR1, AAO, and AUX1/LAX, and an increased activity of BIG.

GAs are crucial plant hormones that regulate a series of activities during plant development [39,40,41,42,43,44]. KEGG analyses further demonstrated a significant enrichment of gibberellin (GA) synthesis and signal transduction with DAPs in the tiller bud and stem node of the *mtn1* mutant (Figure 7, Appendix A). Interestingly, GGPS was down-regulated, and the gibberellin receptor GID1L2 exhibited significant accumulation in *mtn1* mutant (Figure 7, Appendix A). Our study confirmed the findings of previous studies conducted on orchardgrass, indicating a decrease in the expression levels of *DgCPS* and *DgGA2ox*, which are associated with GA synthesis, in multi-tillering cultivars [17]. Similar findings were observed in rice, where overexpression of *OsGID1* led to a decrease in tiller numbers [20]. GGPS catalyzes the synthesis of GGPP from farnesyl diphosphate and isopentenyl diphosphate [40]. This result implies that the biosynthesis of GAs was decreased in the *mtn1* mutant compared to the wild type. We inferred that the *mtn1* mutant may accumulate fewer GAs compared to GGPS, which could account for the differences in tillering ability between the *mtn1* mutant and WT.

BRs are a significant class of hormones involved in regulating cell elongation and differentiation [23,45,46,47,48]. Specifically, CYP92A6 and TCH4 exhibited significant accumulation in the *mtn1* mutant (Figure 7). In this study, the accumulation of CYP92A6 in the *mtn1* mutant may result in an increase in BR content. A similar study has shown that the overexpression of the BR biosynthesis gene *OsHAP1* (*Sterol C*–*22 hydroxylase*) in rice can increase the tiller number [46]. TCH4 is an important target protein downstream of BR signaling and acts as a major component of the plant cell walls, playing a role in cell elongation [49]. We assumed that *mtn1* mutant may accumulate more BRs due to the accumulation of CYP92A6, which could account for the differences in tillering ability between the *mtn1* mutant and WT.

Sugars are also essential for tiller development, serving not only as early signaling substances but also as a source of energy for tiller outgrowth [25,50]. This study found a significant accumulation of many key enzymes involved in starch and sucrose metabolism in the tiller bud of the *mtn1* mutant, including GLU and GBE. Additionally, βAMY, TPS, EG, GBE, and GLU in the stem node were also accumulated (Figure 8, Appendix A). Our study was consistent with previous studies on tiller development in orchardgrass, upland cotton, and wheat [17,32,33]. Those studies also revealed a preferential expression of enzymes related to starch and sucrose metabolism, including TPS, GBE, and GLU in the tiller bud. GLU catalyzes the conversion of disaccharide into D–glucose. TPS catalyzes the conversion of UDP–glucose to 6–phosphate trehalose, and eventually, trehalose is converted into D–glucose [28,51]. The photosynthetic rate in the leaves of the *mtn1* mutant was significantly higher than that of the WT, resulting in the increased transport of photosynthates to the tillers. The increase in the accumulation of these proteins may cause an increase in soluble sugar content in the tiller buds of *mtn1* mutant.

## 4. Materials and Methods

### 4.1. Plant Materials and Experimental Design

The wild type (WT) used in this study was the Yuxi cultivar of centipedegrass (*Eremochloa ophiuroides* (Munro) Hack.), which was bred by our research team (Turfgrass Research Center, Institute of Botany, Jiangsu Province and the Chinese Academy of Sciences). The *mtn1* mutant was generated by our research team using cold plasma mutagenesis technology to induce somatic mutation in the Yuxi callus [52]. The materials were collected from field plots in the turfgrass nursery located at the Institute of Botany, Jiangsu Province, and the Chinese Academy of Sciences in Nanjing, China (118°46′ E, 32°03′ N).

The *mtn1* mutant and WT, both exhibiting consistent growth, were planted in polyethylene plastic buckets (25 cm high, 20 cm in diameter). Each plastic bucket was filled with a 3:1 (*w*/*w*) mixture of 3 kg of soil and sand. The soil exhibited a yellow–brown color and had the following characteristics: pH 6.3, organic matter content of 6.0 g kg^−1^, total nitrogen content of 0.45 g kg^−1^, total phosphorus content of 0.18 g kg^−1^, total potassium content of 8.3 g kg^−1^, available phosphorus content of 4.2 mg kg^−1^, and available potassium content of 107 mg kg^−1^. Before sowing, each bucket was fertilized with 0.95 g of compound fertilizer (N:P:K = 15:15:15, Stanley Agriculture Group Company Limited, Linyi, China). The buckets were placed in a rain-sheltered greenhouse, maintaining a temperature range of 25–35 °C and an air humidity level of 65–80%. Each type of plant was planted in 10 buckets arranged in a randomized complete block design. Regular fertilization and watering were provided to all plants throughout the growth period.

After 2 months of cultivation, the number of tillering, the amount of chlorophyll contents, and the net photosynthetic rate in the third leaf were measured. Simultaneously, tiller buds and the third stem node were randomly collected, frozen in liquid nitrogen, and stored at 80 °C for the measurement of physiological indices and protein/RNA analysis, with 3 replicates for each type of plant.

### 4.2. Physiological Parameters’ Determination

The number of tillers, the amount of chlorophyll contents, and the net photosynthetic rate in the third leaf were measured after two months of cultivation. The tiller buds and the third stem node were randomly collected, frozen in liquid nitrogen, and stored at −80 °C for the measurement of physiological parameters and protein/RNA analysis. Three replicates were performed for each type of plant. The number of first tillers (tillers on the main stolon) and secondary tillers (tillers on the first tiller) were measured after 2 months of planting. Ten individuals were randomly selected from each sample.

The chlorophyll contents were determined using the method of 95% ethanol extraction [53]. Briefly, fresh leaf samples weighing 0.2 g were ground with 95% ethanol to form a homogenate. The homogenate was then filtered into a brown volumetric bottle, maintaining a constant volume of 25 mL. The absorbance at wavelengths of 665 nm, 649 nm, and 470 nm was determined using an Ultrospec 3300 Pro spectrophotometer (Amersham Biosciences, Uppsala, Sweden). The chlorophyll content was calculated according to the following formula: chlorophyll content (mg g^−1^) = (chlorophyll concentration × extraction liquid volume × dilution factor)/sample fresh weight. The net photosynthetic rate was measured using the LI–6800 photosynthesis measurement system (Li–COR, Lincoln, NE, USA) equipped with a standard leaf chamber. The starch, soluble sugar, and sucrose contents were determined using the anthrone colorimetry test, as described previously [54,55]. Briefly, dry tiller bud/stem node samples weighing 0.1 g were ground with 5 mL of 80% ethanol and centrifuged at 3000× *g* for 10 min. For the starch content assay, the centrifuged pellet was redissolved in 3 mL of distilled water and incubated in a 100 °C water bath for 10 min. Then, 2 mL of 1.1% (*v*/*v*) HCl was added to enhance the degradation of starch into soluble sugar. The mixture was incubated in a 100 °C water bath for 30 min. The absorbance at 625 nm was determined using an Ultrospec 3300 Pro spectrophotometer (Amersham Biosciences, Uppsala, Sweden). The starch content was determined using the following formula: starch content (mg g^−1^) = starch content in standard curve/weight of sample. To determine the soluble sugar content, the supernatants were mixed with a five-fold volume of 1% (*m*/*v*) anthrone dissolved in H_2_SO_4_. The absorbance at 620 nm was determined using an Ultrospec 3300 Pro spectrophotometer (Amersham Biosciences, Uppsala, Sweden). The soluble sugar content was calculated based on the sugar content obtained from the standard cure. For the sucrose content assay, the collection was held at 80 °C for 45 min. A total volume of 200 μL of sodium hydroxide (2 mol L^−1^) was added into 0.4 mL of the extract. The mixture was held in a 100 °C water bath for 5 min. The mixture was then reacted with 2.8 mL of 30% hydrochloric acid and 1% resorcinol solution at 80 °C for 10 min. The absorbance at 480 nm was determined using an Ultrospec 3300 Pro spectrophotometer (Amersham Biosciences, Uppsala, Sweden). Sucrose content was calculated according to the following formula: sucrose content (mg g^−1^) = sucrose content in standard curve/weight of sample.

### 4.3. Protein Extraction and Trypsin Digestion

The protein was extracted from tiller buds, and stem nodes were extracted using a modified Tris-phenol method as previously described [56]. Briefly, 1 g of tiller bud and 1 g of stem node samples were placed in a shaking tube and an appropriate volume of protein lysis solution was added to the samples. The samples were then shaken using a tissue grinder, and the lysis was performed on ice. The supernatant was collected by centrifugation. The protein content was determined using the bicinchoninic acid (BCA) method. Subsequently, 100 μg of protein from each sample was mixed with lysis solution and 100 mM triethylammonium bicarbonate buffer (TEAB) for overnight trypsin digestion at 37 °C. The peptide segments were then dried and redissolved in 0.1% trifluoroacetic acid and quantified using the ThermoFisher Scientific peptide quantification kit (ThermoFisher Scientific, Waltham, MA, USA).

### 4.4. Liquid Chromatography Tandem Mass Spectrometry (LC-MS/MS) Analysis

The peptide segments were dissolved in the buffer solution and separated using the EASY–nLC1000 liquid phase system with a C18 column (75 μm × 25 cm, ThermoFisher Scientific, Waltham, MA, USA). Mobile phase A comprised 2% acetonitrile and 0.1% formic acid, while mobile phase B comprised 80% acetonitrile and 0.1% formic acid. The gradient for the mobile phase was as follows: 0–6% for 0–2 min; 6–23% for 2–105 min; 23–29% for 105–130 min; 29–38% for 130–147 min; 38–48% for 147–148 min; and 48–100% for 148–149 min. The mass spectrometry analysis was conducted using tims TOF Pro2 (Bruker corporation, Bruker, Germany) with a first-level mass resolution of 70,000 and a second-level resolution of 17,500.

### 4.5. Protein Identification and Bioinformatics Analysis

The raw MS spectra were analyzed using ProteomeDiscovererTM Software 2.2 (ThermoFisher Scientific, Waltham, MA, USA) and the centipedegrass database [23]. The following parameters were used: cysteine alkylation (iodoacetamide), variable modification (methionine oxidation), enzyme (trypsin), bias correction (true), background correction (true), protein mass (unrestricted), unused score (≥1.3), confidence (≥95%), unique peptides (≥1), and unique peptide (≥1). Statistical analysis was conducted to examine differential protein expression using the abundance information obtained from the database search. MaxQuant software 2.0 (Max Planck Institute of Biochemistry, Martinsried, Germany) was utilized for label-free quantitation (LFQ) to identify peptides and proteins and to determine their relative abundance [57]. The Plant–mPLoc server 2.0 (Shanghai Jiao Tong University, Shanghai, China) was employed to predict the subcellular locations of the DAPs [58]. Peptide and protein identifications were accepted if they exceeded a 95% and 99% probability threshold, respectively. Proteins with less than two peptides per protein were excluded, and proteins sharing redundant peptides were grouped together. The Scaffold Local FDR algorithm 4.6.2 and Protein Prophet algorithm (http://sf.net/projects/proteinprophet/ (accessed on 25 March 2024), Proteome Software Incorporated, Portland, OR, USA) were used to calculate the probabilities of peptides and proteins, respectively [59].

Student’s *t*-test was used to calculate the significance of differences between samples. Proteins exhibiting with *p* < 0.05 and a fold change ≥1.5 or ≤0.5 were considered as differentially abundant proteins (DAPs) using one-way analysis of variance (ANOVA) and Student–Newman–Keuls tests using SPSS 16.0 software. Gene Ontology (GO) and Kyoto Encyclopedia of Genes and Genomes (KEGGs) annotation information for the identified protein species was obtained by performing BLAST searches against the GO database ((http://geneontology.org/ (accessed on 25 March 2024), Gene Ontology Consortium, Bethesda, MD, USA) and the KEGG database (http://www.genome.jp/kegg// (accessed on 25 March 2024), Bioinformatics Research Center, Institute of Chemistry, Kyoto University, Kyoto, Japan), respectively [60,61]. The software Goatools ((https://github.com/tanghaibao/goatools (accessed on 25 March 2024), Haibao Tang, San Franciso, CA, USA) was employed for conducting GO enrichment analysis and KEGG analysis to determine the significance of differences; Fisher’s exact test was applied for verification. A corrected *p*-value threshold of ≤0.05 was considered indicative of significant enrichment in both GO function and KEGG pathway. The MapMan software (https://mapman.gabipd.org/home (accessed on 25 March 2024), Max Planck Institute for Molecular Plant Physiology, Potsdam, Germany) was used to visualize the metabolic and regulatory pathways. KEGG analysis was employed to screen for DAPs involved in plant hormone signal transduction and starch and sucrose metabolism.

### 4.6. qRT-PCR

The total RNA was extracted from tiller buds and stem nodes using the RNA prep Pure Plant Kit (Takara Group Company, Osaka, Japan). Primers for the DAPs were designed using Primer 5.0 software (Appendix A). Quantitative real-time PCR (qRT-PCR) was performed using the Bio-Rad CFX Connect PCR Detection System (Bio-Rad, Hercules, CA, USA) [59]. Each sample was replicated three times.

### 4.7. Data Statistics and Analysis

The morphological, physiological, and qRT-PCR data are presented as means ± standard errors. Data visualization was carried out using Excel 2021 and PPT 2021 (Microsoft Corporation, Redmond, WA, USA). Principal component analysis (PCA) was conducted on the identified proteins using the ggplot2 package in R language software (v 3.5.0, R Core Team, Copenhagen Business School, Frederiksberg, Denmark). The Student–Newman–Keuls test and Tukey’s multiple-comparison test were performed using SPSS 16.0 software (International Business Machines Corporation, Armonk, NY, USA) to analyze variations in the mean of two samples and multiple samples, respectively.

## 5. Conclusions

In summary, this study conducted an analysis of DAPs in the tiller buds and stem nodes of the *mtn1* mutant, identifying 438 and 589 DAPs, respectively. The results revealed that plant hormones’ signaling pathways, as well as pathways and interactions related to sucrose and starch metabolism, may be the primary factors contributing to the high tillering observed in the *mtn1* mutant. Results of this study provide crucial information for future functional studies on specific proteins that regulate tiller development in centipedegrass and other clonal plants. The molecular mechanism underlying the regulation of tiller development by functional proteins related to plant hormones and sugars will be investigated in future research.

## Figures and Tables

**Figure 1 plants-13-01028-f001:**
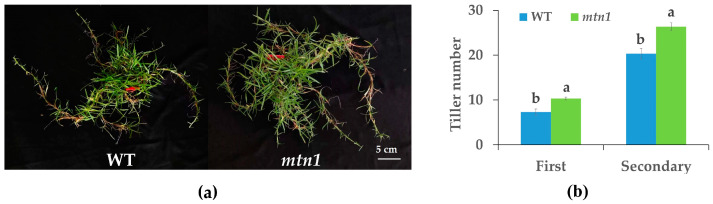
Comparison of the tillering phenotype between the *mtn1* mutant (right) and WT (left) in 60-day-old plants (**a**); the first and secondary tiller numbers of *mtn1* mutant and WT (**b**). Data are shown as mean ± standard error. Different lowercases indicate significant differences determined by Student–Newman–Keuls tests (*p* < 0.05).

**Figure 2 plants-13-01028-f002:**
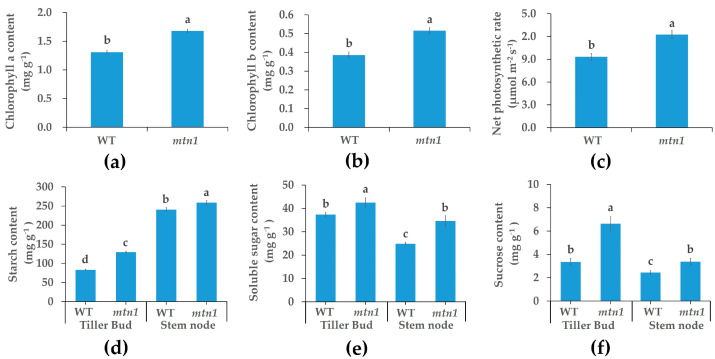
Physiological differences between *mtn1* mutant and WT in centipedegrass. (**a**) Chlorophyll a content, (**b**) chlorophyll b content, (**c**) net photosynthetic rate, (**d**) starch content, (**e**) soluble sugar content, and (**f**) sucrose contents in the tiller buds and stem nodes of *mtn1* mutant and WT. Data are shown as mean ± standard error. Different lowercases indicate significant differences determined by Student–Newman–Keuls tests (*p* < 0.05).

**Figure 3 plants-13-01028-f003:**
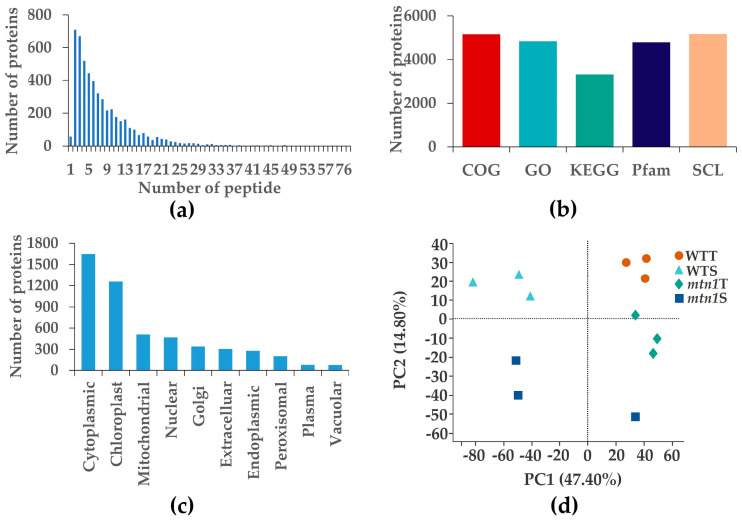
Overview of the proteome data. (**a**) Peptide distribution in identified proteins, with the horizontal axis representing the range of peptides covering each protein, and the vertical axis representing the count. (**b**) Protein annotation results, including Clusters of Orthologous Groups (COGs), Gene Ontology (GO), Kyoto Encyclopedia of Genes and Genomes (KEGGs), protein family database (Pfm), and subcellular location (SCL). (**c**) Prediction of subcellular localization in identified proteins. (**d**) Principal component analysis (PCA); WTT: WT tiller bud, WTS: WT stem node, *mtn1*T: *mtn1* tiller bud, *mtn1*S: *mtn1* stem node.

**Figure 4 plants-13-01028-f004:**
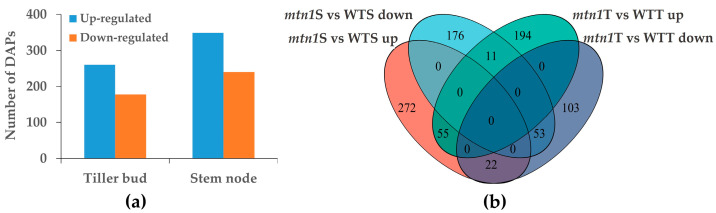
The number of DAPs in tiller buds and stem nodes (**a**); Venn diagram of differentially expressed proteins (**b**). *mtn1*S: *mtn1* stem node, WTS: WT stem node, *mtn1*T: *mtn1* tiller bud, WTT: WT tiller bud.

**Figure 5 plants-13-01028-f005:**
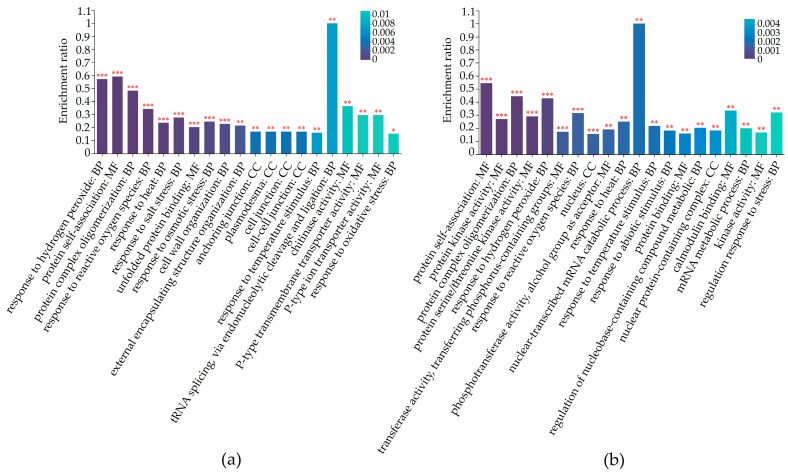
GO analysis of the functional classification of DAPs in in tiller buds (**a**) and stem nodes (**b**). The color bars represent the significance of GO enrichment. The darker the default color, the more significant the enrichment of the GO pathway. Markers with a significance level of *p* < 0.001 are represented by ***, *p* < 0.01 are represented by **, and *p* < 0.05 are represented by *. *p* ≤ 0.05 was considered indicative of significant enrichment in GO function.

**Figure 6 plants-13-01028-f006:**
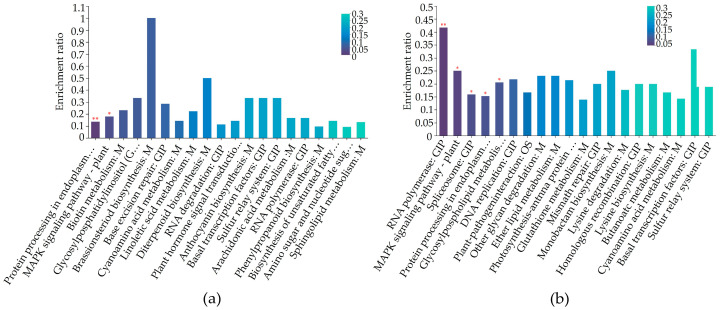
KEGG pathway enrichment of DEPs in tiller buds (**a**) and stem nodes (**b**). The color bars indicate the significance of KEGG enrichment. The darker the default color, the more significant the enrichment of the KEGG pathway. Markers with a significance level of *p* < 0.01 are represented by **, and *p* < 0.05 are represented by *. *p* ≤ 0.05 was considered indicative of significant enrichment in KEGG pathway.

**Figure 7 plants-13-01028-f007:**
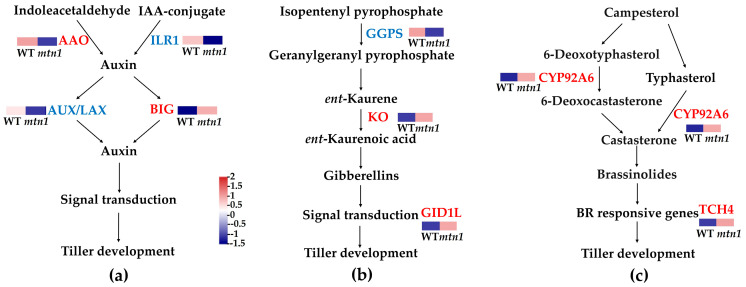
DAPs involved in plant hormone signal transduction pathways. (**a**) Pathway involved in auxin signal transduction, (**b**) pathway involved in GA signal transduction, (**c**) pathway involved in BR signal transduction. The expression data are the TPM values of the samples; pink indicates up-regulated expression, and blue indicates down-regulated expression.

**Figure 8 plants-13-01028-f008:**
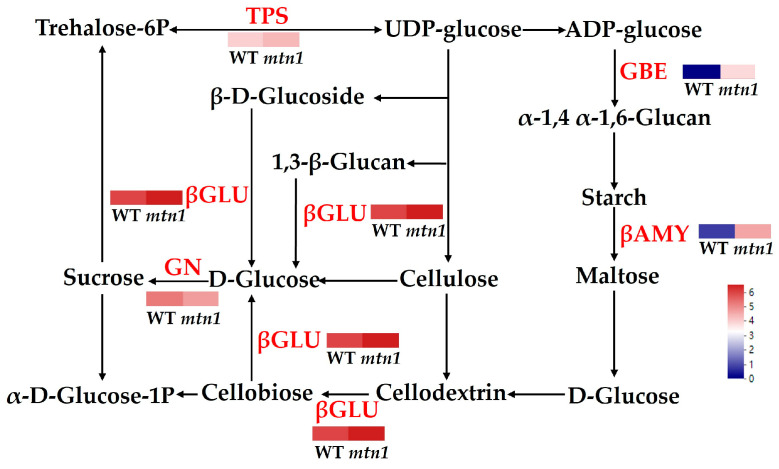
DAPs involved in starch and sucrose metabolism. The expression data are the TPM values of the samples; red and pink indicate up-regulated expression, and blue indicates down-regulated expression.

**Figure 9 plants-13-01028-f009:**
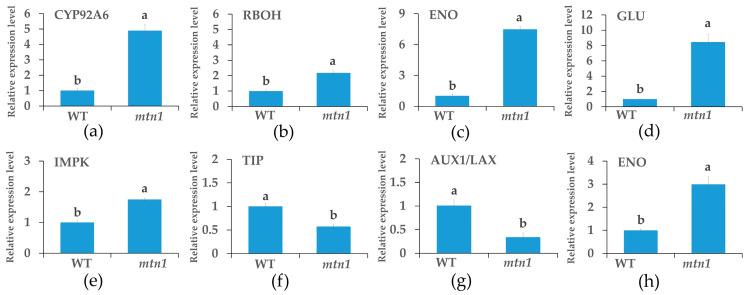
qRT-PCR analysis of mRNA expression levels of selected genes encoding proteins that differentially accumulated in tiller bud (**a**–**d**) and stem node (**e**–**h**). CYP926A: typhasterol/6–deoxotyphasterol 2alpha–hydroxylase, RBOH: respiratory burst oxidase, ENO: enolase, GLU: beta-glucosidase, IPMK: inositol-polyphosphate multikinase, TIP: aquaporin, AUX1/LAX: auxin influx carrier (AUX1/LAX family). Eo–*ACTIN* was used as an internal control for normalization. Data are shown as mean ± standard error. Different lowercases indicate significant differences determined by Student–Newman–Keuls tests (*p* < 0.05).

**Figure 10 plants-13-01028-f010:**
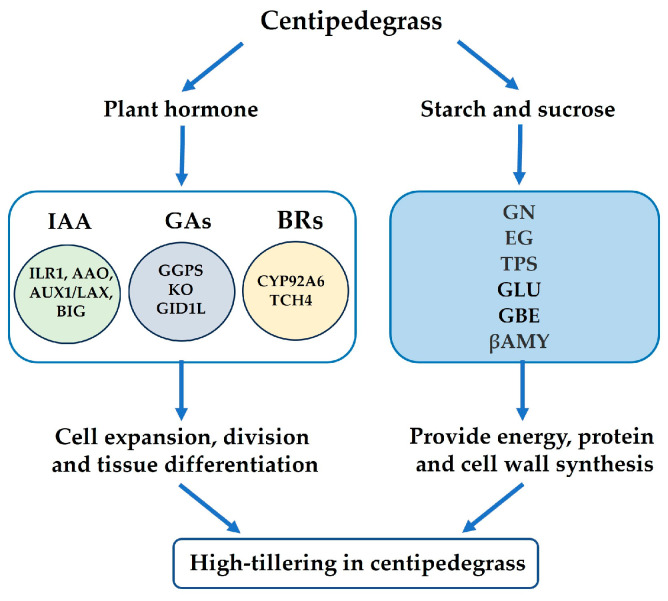
A hypothetical interaction network between auxin, gibberellins (GAs), brassinosteroids (BRs) and sugars is proposed to regulate high tillering in the *mtn1* mutant in centipedegrass.

## Data Availability

The mass spectrometry proteomics data have been deposited to the ProteomeXchange Consortium via the PRIDE partner repository with the dataset identifier PXD047559.

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
