# Peer review of "Physiological and Proteomic Analyses of mtn1 Mutant Reveal Key Players in Centipedegrass Tiller Development"

_plants, 2024, doi:10.3390/plants13071028_

Round 1

Reviewer 1 Report

Comments and Suggestions for Authors

In my opinion, the present physiological and proteomic study of the mtn1 mutant and its wild type in improving tillering in centipedegrass is interesting. The study seems to be well developed with conclusive results. However, important improvements need to be made before publication. At the level of the introduction and discussion, the background needs to be improved, as only 40 references are cited at the end of the manuscript. The other notable aspect is that the methodological information needs to be improved, following the logical order of the results and with all the information. The results could be improved at the level of the format of the figures and legends, and a clearer description of the differences at the quantitative level. Detailed information can be found in the attached file.

Comments on the Quality of English Language

In my opinion, moderate editing of the English language is required.

Author Response

Responses to Reviewer 1

Dear reviewer:

Thank you for your valuable comments on this manuscript. During the revision process, we addressed the reviewer's comments and suggestions by implementing necessary changes in a systematic manner. Firstly, we enhanced the background section by incorporating additional references. Secondly, we revised and enhanced the methodological information, ensuring its alignment with the logical order of the results and incorporating all relevant details. Thirdly, we made modifications to the results section. Finally, we have rewritten the discussion section and made additional modifications to the entire manuscript based on the reviewer's suggestions.

Detailed remarks:

  1. Title: I think you can omit including (more tillering number 1) this here

Response: Thank you for your valuable comment and suggestion. In response, I have removed “more tillering number 1 from the title and highlighted the changes in the revised version in line 2.

  1. Line 14: You could also include relevant results in a quantitative way.

Response: Thank you very much for this excellent suggestion. As per your suggestion, we have incorporated the relevant results in a quantitative manner. Changes were highlighted in the revised version in line 18-23.

  1. Line 28: You should include more background bibliographic references that include the most significant advances at the molecular level in relation to the parameters analyzed in this study.

Response: Thank you very much for this good suggestion. We have incorporated the latest molecular advancements related to the parameters investigated in this study. Changes were highlighted in the revised version in line 52-79.

  1. Line 42: add reference.

Response: Thank you very much for this comment. I have included the appropriate reference. Changes were highlighted in the revised version in line 52.

  1. Line 59-65: This belongs to the discussion. After having discussed the prior existing knowledge relating to this study and the justification for its need, here you should indicate the main objectives of this study, preferably point by point.

Response: Thank you very much for this good comment. We have modified these sentences as follows: The study aimed to investigate the molecular regulatory mechanism of tiller development in centipedegrass by evaluating comparative phenotypic, physiological, and proteomic changes between the mtn1 mutant and wild type. Specifically, the objectives were: (1) to characterize physiological traits, such as photosynthetic activity, starch and soluble sugar contents, in the mtn1 mutant and wild type; (2) to identify differentially accumulated proteins (DAPs) that contribute to the high tiller numbers in the mtn1 mutant; and (3) to analyze the mechanisms underlying tiller development in centipedegrass by integrating physiological and proteomic changes. We have highlighted changes in the revised version in line 87-94.

  1. Line 70: Please, include "first tillering" and "secondary tillering" in Material and Methods, explaining them. Include also age-days in Figure 1.

Response: Thanks for this good suggestion. We have explained the "first tillering" and "secondary tillering" as follows: tillers on the main stolon and tillers on the first tiller. Additionally, we included the age-days in Figure 1. The revisions were indicated and highlighted in the updated version in line 116 and 443.

  1. Line 77: 'significantly higher levels'... please, try to include also differences in a quantitative way (for example, n-fold or %). Extensible to the rest of the manuscript.

Response: Thanks for your valuable. We have made modifications to the provided information and presented the results in a quantitative manner. Changes were highlighted in the revised version in line 109-110, 119-127.

  1. Line 84: Please make the letters on the graph larger and according to the MDPI format. Include units of measurement also in the legend, correct formatting errors. Extendable to the rest of the figures.

Response: Thanks for this good comment. We have followed your suggestion and redrew the figures. Changes has been highlighted in the revised version.

  1. Line 86: Error bars represent SE. Data are shown as mean ± standard error.

Response: Thanks for this good comment. We replaced the statement “Error bars represent SE” with “Data are shown as mean ± standard error” in related figure legends. Changes were highlighted in the revised version.

  1. Line 88: The results must follow the logical order of the Materials and Methods. Clearly explain this section in MM following a logical order in both MM and Results. Also review the rest of the sections.

Response: We sincerely appreciate your valuable suggestion. This section has been comprehensively described in the Materials and Methods section, as well as other relevant sections. The revised version clearly indicates the changes made (line 509-521).

  1. Line 90: LFQ proteomic analysis

Response: Thanks for this good comment. In response to your feedback, we have included the full name of label-free quantitation (LFQ) proteomic analysis. Changes were highlighted in the revised version in line 140.

  1. Line 102-104: Please, improve the legend. It must be possible to study the graphs with the information included in the legend.

Response: Thanks for this good comment. We changed this legend as follows:  Figure 3. Overview of the proteome data. A: Peptide distribution in identified proteins, with the horizontal axis representing the range of peptides covering each protein, and the vertical axis representing the count. B: Protein annotation results including Usters of Orthologous Groups (COG), Gene Ontology (GO), Kyoto Encyclopedia of Genes and Genomes (KEGG), protein family database (Pfm), and subcellular location (Subcell-Location). C: Prediction of subcellular localization in identified proteins. D: Principal component analysis (PCA). Changes were highlighted in the revised version in line 158-163.

  1. Be careful with the format, the document must be perfect for possible publication.

Response: Thank you very much for this good suggestion. We have modified the format of this section and indicated the revisions in the updated version in 182-194.

  1. Line 129: This information is not included in the statistics section of Materials and Methods.

Response: Thank you very much for this good suggestion. The markers were mislabeled and subsequently removed. Changes were highlighted in the revised version in line 208.

  1. Line 132: in in

Response: Thank you very much for this comment. We have removed an unnecessary word. Changes were highlighted in the revised version in line 212.

  1. Line 142: Figure 6

Response: Thanks for this comment. We have made modifications to the format of "Figure 6" as well as other figures. Changes were highlighted in the revised version in line 225, 251, 278, 291.

  1. Line 146: Please, include this information in MM, also for 2.7.

Response: Thank you for your comment. We added how to conduct this part in Materials and Methods section as follows: The MapMan software (Max Planck Institute for Molecular Plant Physiology, Potsdam, Germany) was used to visualize the metabolic and regulatory pathways. KEGG analysis was employed to screen for DAPs involved in plant hormone signal transduction, starch and sucrose metabolism. Changes were highlighted in the revised version in 530-533.

  1. Line 194: Please, include references.

Response: Thanks for this good comment. We added references in this part. Changes were highlighted in the revised version in line 299 and 302.

  1. Line 200: This should be discussed with bibliographic information.

Response: Thanks for this good comment. We modified these sentences as follows:  This study conducted a quantitative physiological and proteomic analysis of the tiller buds and stem nodes of the mtn1 mutant and WT. The results showed an increase in chlorophyll content, net photosynthetic rate in leaves, and starch, soluble sugar, and sucrose contents in the tiller bud and stem node of the mtn1 mutant (Figure 2). The IAA and GA contents in the tiller bud of the mtn1 mutant were reduced [14]. A total of 438 and 589 DEPs DAPs were identified in the tiller buds and stem nodes, respectively (Figure 4A). The enriched proteins in the mtn1 mutant were found to be involved in plant hormone signal transduction pathways and starch and sucrose metabolic pathways, suggesting the potential regulatory roles of plant hormones and sugars in centipedegrass tiller development. These results provide insights into tiller development in clonal plants. Modifications were highlighted in the revised version in line 302-312.

  1. Line 204: Please, rewrite. The text must follow the logical order of the results. After indicating a result, or prior to a brief introduction, you must justify the differences observed with the information existing in the bibliography, but in a connected and logical manner. This extends to the rest of the discussion, which should be more complete, since it is very brief.

Response: Thank you very much for this good suggestion. We have revised the discussion section as you suggested. Changes were highlighted in the revised version in line 307-358.

  1. Line 255: Please also include the scientific name and author here, and also include the origin of the material studied.

Response: Thanks for this good suggestion. The scientific name, author, and origin of the studied material were included as follows: The wild type (WT) used in this study was the Yuxi cultivar of centipedegrass (Eremochloa ophiuroides (Munro) Hack.), which was bred by our research team (Turfgrass Research Center, Institute of Botany, Jiangsu Province and the Chinese Academy of Sciences). The materials were collected from field plots in the turfgrass nursery located at the Institute of Botany, Jiangsu Province, and the Chinese Academy of Sciences in Nanjing, China (118°46′E, 32°03′N). The revised version highlights the changes made in line 410-416.

  1. Line 256: Please include a reference explaining the method.

Response: Thanks for this good comment. We have included a patent, invented by our team, which describes the technique of cold plasma mutagenesis breeding in detail. The revised version highlights the changes in line 414.

  1. Line 259: The soil was yellow-brown. Texture?

Response: Thanks for this good comment. The soil type is characterized as yellow-brown.

  1. Line 263: Please include the manufacturer, city and country.

Response: Thank you for this suggestion. We added full information of this fertilizer as follows: compound fertilizer (N: P: K = 15: 15: 15, Stanley Agriculture Group Company Limited, Linyi, China). Changes were highlighted in the revised version in line 427.

  1. Line 267: I think you should integrate this information in the following points.

Response: Thank you for this good suggestion. As per your suggestion, we have incorporated this information into the section on determination of physiological parameters. The revised version contains highlighted changes in line 432-471.

  1. Line 272: index parameters?

Response: Thanks for this good comment. The index has been substituted with parameters. The revised version contains highlighted change in line 438.

  1. Line 275: Please, include a reference or describe it briefly.

Response: Thanks for this good review. We have included a reference [53] that is directly relevant to the proposed method. Changes were highlighted in the revised version in line 454.

  1. Line 277: You can describe it briefly. You should include the units of measurement of the analyzed parameters, extendable to the rest of the manuscript.

Response: Thanks for this good suggestion. We provided a concise description of the method used to determine various physiological parameters. Changes were highlighted in the revised version in line 446-471.

  1. Line 288: Please, include also city and country.

Response: Thanks for this good comment. We included information on the manufacturer, city, and country of origin for the reagent. Changes were highlighted in the revised version.

  1. Line 289: You should include also the complete name.

Response: Thanks for this good comment. We added the full information as follows: Liquid chromatography tandem mass spectrometry (LC-MS/MS) analysis. Changes were highlighted in the revised version in line 485.

  1. Line 291: Please include the brand of the measurement equipment, city and country. Extendable to the rest of the manuscript.

Response: Thanks for this good comment. Throughout the article, we included information about the measurement equipment, as well as the city and country. Changes were highlighted in the revised version.

  1. Line 300: Same previous comments, please include the manufacturer name, city and country for all the software.

Response: Thanks for this good comment. We added the related information for all software. Changes were highlighted in the revised version.

  1. Line 301: This paragraph is a little confusing. Can you write it in a linear way?

Response: Thanks for this good comment. We rewrote this paragraph as follows: The parameters were set as follows: The parameters were set as followsused: cysteine alkylation (iodoacetamide), variable modification (methionine oxidation), enzyme (trypsin), bias correction (true), back-ground correction (true), protein mass (unrestricted), unused score (≥ 1.3), confidence (≥ 95%), unique peptides (≥ 1), and unique peptide (≥ 1). Changes were highlighted in the revised version in line 500-504.

  1. Line 320: Please, include also information concerning normality and homoscedasticity of the data.

Response: Thanks for your suggestion. As you suggested, we added this information as follows: The ANOVA statistical analysis was performed using SPSS 16.0 software (Internation-al Business Machines Corporation, Armonk, USA). Data visualization was carried out using Excel 2021 and PPT 2021 (Microsoft Corporation, Redmond, USA). Duncan's test was utilized for variance analysis and to determine significant differences (p < 0.05). Changes were highlighted in the revised version in line 545-549.

  1. Line 325: The conclusions seem incomplete to me. It should also include information about the study's projection.

Response: Thanks for this good comment. In the conclusion section, we added the projection of this study as follows: The molecular mechanism underlying the regulation of tiller development by functional proteins related to plant hormones and sugars will be investigated in future research. Changes were highlighted in the revised version in line 568-570.

  1. Line 330: Please, include this information in Material and Methods.

Response: Thanks for this good suggestion. We added the description of this information in Material and Methods section as follows: The MapMan software (Max Planck Institute for Molecular Plant Physiology, Potsdam, Germany) was used to visualize the metabolic and regulatory pathways. KEGG analysis was employed to screen for DAPs involved in plant hormone signal transduction, starch and sucrose metabolism. Changes were highlighted in the revised version in line 529-533.

  1. Line 353: The background needs to be improved.

Response: Thanks for this good comment. We have expanded the background information and enhanced this section. Changes were highlighted in the revised version in 36-94.

Sincerely,

Ling Li

Jiangsu Province and Chinese Academy of Sciences

Zhongshanmenwai Qianhuahoucun, No. 1, Nanjing 210014, China

Reviewer 2 Report

Comments and Suggestions for Authors

Dear Authors,

Reviewer comments plants-2903077

The manuscript entitled „Physiological and proteomic analysis of a mtn1 (more tillering number 1) mutant reveal key players in centipedegrass tillering development“ represents a useful study focused on a comparison of centipedegrass (Eremochloa ophiuroides) wild-type (WT) and more tillering number 1 (mtn1) mutant revealing enhanced tillering at physiological (plant growth characteristics, chlorophyll content, net photosynthesis rate) and proteomic (differentially abundant proteins between WT and mtn1) levels. A comparison of both WT and mtn1 revealed enhanced net photosynthesis rate and altered hormone biosynthesis, transport and signal transduction leading to enhanced soluble sugar content which underlie enhanced tillering in mtn1 with respect to WT.

I can recommend the manuscript for publication in Plants. However, I have a few major comments on the present version of the manuscript.

1/ In Introduction, basic information on mtn1 mutant has to be given. What is known about the structure and functions of mtn1 gene?

2/ In Materials and methods, Plant materials section, the sources of both centipedegrass WT cv. Yuxi and mtn1 mutant have to be given. From which institution were they obtained?

In LC-MS/MS analysis, the kind of LC-MS/MS systém (EASY-nLC1200 liquid phase systém) has to eb specified by providing the information on the manufacturer (company).

3/ Results: QRT-PCR cannot be considered a validation of proteomic results since qRT-PCR determines transcript levels which can reveal differential dynamics from the proteins. qRT-PCR can be considerd only as a complementation to proteomic data but not their validation. For validation of LC-MS/MS data, another kind of proteomic approach, e.g., determination of selected DAPs identified by LC-MS/MS by immunoblot analysis should be employed.

In Figure 1A, an appropriate scale bar has to eb added to the photos of both WT and mtn1.

4/ Terminology:

Use either the term „tillering stage“ or „tiller development“ but not „tillering development“ including the manuscript title: „Physiological and proteomic analysis of a mtn1 (more tillering number 1) mutant reveal key players in centipedegrass tiller development“

Use the term „DAPs“, i.e., „differentially abundant proteins“ instead of „DEPs“, i.e., „differentially expressed proteins“ since the differences in protein relative abundance always represent a result between protein biosynthesis („protein expression“) and protein degradation.

Line 321: Correct the term „phycological data“ to „physiological data“ since „phycological“ means associated with algae.

5/ In Conclusion, a new figure (Figure 10) providing a model scheme comparing the molecular mechanisms involved in tiller development between centipedegrass WT and mtn1 mutant based on physiological and proteomic results of the present study.

6/ Formal comments on the etxt related to English language and style:

Introduction, line 53: Add a comma following the word „Recently,….“

Results, Line 78: Modify the word form „In consistent“ to „In consistence“ in the statement: „In consistence with the chlorophyll contents,…“

Discussion, line 215: Modify the verb form „served“ to the present form „serves“ in the statement: „The BIG protein serves as an auxin efflux carrier…“

Line 220: Use the word form „enhanced auxin transport“  instead of „more auxin transport.“

Line 226: Correct the spelling in „GAs“ not „Gas“ for gibberellins.

Line 251: Modify the word form ůaccumulated“ to „accumulation“ in the statement: „The increase in the accumulation of these proteins may cause an increase in soluble sugar content in tiller buds of mtn1 mutant.“

Line 321: Correct the term „phycological data“ to „physiological data“ since „phycological“ means associated with algae.

Final recommendation: Reconsider after a major revision.

Comments on the Quality of English Language

Dear Authors,

Reviewer comments plants-2903077

The manuscript entitled „Physiological and proteomic analysis of a mtn1 (more tillering number 1) mutant reveal key players in centipedegrass tillering development“ represents a useful study focused on a comparison of centipedegrass (Eremochloa ophiuroides) wild-type (WT) and more tillering number 1 (mtn1) mutant revealing enhanced tillering at physiological (plant growth characteristics, chlorophyll content, net photosynthesis rate) and proteomic (differentially abundant proteins between WT and mtn1) levels. A comparison of both WT and mtn1 revealed enhanced net photosynthesis rate and altered hormone biosynthesis, transport and signal transduction leading to enhanced soluble sugar content which underlie enhanced tillering in mtn1 with respect to WT.

I can recommend the manuscript for publication in Plants. However, I have a few major comments on the present version of the manuscript.

1/ In Introduction, basic information on mtn1 mutant has to be given. What is known about the structure and functions of mtn1 gene?

2/ In Materials and methods, Plant materials section, the sources of both centipedegrass WT cv. Yuxi and mtn1 mutant have to be given. From which institution were they obtained?

In LC-MS/MS analysis, the kind of LC-MS/MS systém (EASY-nLC1200 liquid phase systém) has to eb specified by providing the information on the manufacturer (company).

3/ Results: QRT-PCR cannot be considered a validation of proteomic results since qRT-PCR determines transcript levels which can reveal differential dynamics from the proteins. qRT-PCR can be considerd only as a complementation to proteomic data but not their validation. For validation of LC-MS/MS data, another kind of proteomic approach, e.g., determination of selected DAPs identified by LC-MS/MS by immunoblot analysis should be employed.

In Figure 1A, an appropriate scale bar has to eb added to the photos of both WT and mtn1.

4/ Terminology:

Use either the term „tillering stage“ or „tiller development“ but not „tillering development“ including the manuscript title: „Physiological and proteomic analysis of a mtn1 (more tillering number 1) mutant reveal key players in centipedegrass tiller development“

Use the term „DAPs“, i.e., „differentially abundant proteins“ instead of „DEPs“, i.e., „differentially expressed proteins“ since the differences in protein relative abundance always represent a result between protein biosynthesis („protein expression“) and protein degradation.

Line 321: Correct the term „phycological data“ to „physiological data“ since „phycological“ means associated with algae.

5/ In Conclusion, a new figure (Figure 10) providing a model scheme comparing the molecular mechanisms involved in tiller development between centipedegrass WT and mtn1 mutant based on physiological and proteomic results of the present study.

6/ Formal comments on the etxt related to English language and style:

Introduction, line 53: Add a comma following the word „Recently,….“

Results, Line 78: Modify the word form „In consistent“ to „In consistence“ in the statement: „In consistence with the chlorophyll contents,…“

Discussion, line 215: Modify the verb form „served“ to the present form „serves“ in the statement: „The BIG protein serves as an auxin efflux carrier…“

Line 220: Use the word form „enhanced auxin transport“  instead of „more auxin transport.“

Line 226: Correct the spelling in „GAs“ not „Gas“ for gibberellins.

Line 251: Modify the word form ůaccumulated“ to „accumulation“ in the statement: „The increase in the accumulation of these proteins may cause an increase in soluble sugar content in tiller buds of mtn1 mutant.“

Line 321: Correct the term „phycological data“ to „physiological data“ since „phycological“ means associated with algae.

Final recommendation: Reconsider after a major revision.

Author Response

Responses to Reviewer 2

Dear reviewer:

Thank you for your valuable comments on this manuscript. Thank you very much for your good comments on this manuscript. The manuscript has been thoroughly revised based on the reviewer's comments and suggestions, addressing each point in detail.

Detailed remarks:

  1. In Introduction, basic information on mtn1 mutant has to be given. What is known about the structure and functions of mtn1 gene?

Response: Thank you very much for this comment and good suggestion. We added the basic information on mtn1 mutant as follows: The mtn1 mutant was generated by our research team using cold plasma mutagenesis technology to induce somatic mutation in the Yuxi callus. The EoMTN1 gene encodes ABCB11, which belongs to the ATP-binding cassette protein subfamily B (ABCB) transporter family [14]. We have highlighted changes in the revised version in line 84-87.

  1. In Materials and methods, Plant materials section, the sources of both centipedegrass WT cv. Yuxi and mtn1 mutant have to be given. From which institution were they obtained?

Response: Thanks for this good suggestion. We added the source of both plant materials as follows: The wild type (WT) used in this study was the Yuxi cultivar of centipedegrass (Eremochloa ophiuroides (Munro) Hack.), which was bred by our research team (Turfgrass Research Center, Institute of Botany, Jiangsu Province and the Chinese Academy of Sciences). The mtn1 mutant was generated by our research team using cold plasma mutagenesis technology to induce somatic mutation in the Yuxi callus [51]. The materials were collected from field plots in the turfgrass nursery located at the Institute of Botany, Jiangsu Province, and the Chinese Academy of Sciences in Nanjing, China (118°46′E, 32°03′N). Changes were highlighted in the revised version in line 410-416.

  1. In LC-MS/MS analysis, the kind of LC-MS/MS system (EASY-nLC1000 liquid phase system) has to be specified by providing the information on the manufacturer (company).

Response: Thanks for this good suggestion. We provided the information on the manufacturer as follows: The peptide segments were dissolved in a buffer and separated using the EASY-nLC1200 nLC1000 liquid phase system with a C18 column (75μm × 25cm, ThermoFisher Scientific, Waltham, USA). The Mass spectrometry analysis was performed conducted using tims TOF Pro2 (Bruker corporation, Bruker, Germany) with a first-level mass resolution of 70000 and a second-level resolution of 17500. Changes were highlighted in the revised version in line 487, 495.

  1. Results: QRT-PCR cannot be considered a validation of proteomic results since qRT-PCR determines transcript levels which can reveal differential dynamics from the proteins. qRT-PCR can be considerd only as a complementation to proteomic data but not their validation. For validation of LC-MS/MS data, another kind of proteomic approach, e.g., determination of selected DAPs identified by LC-MS/MS by immunoblot analysis should be employed.

Response: Thanks for this good suggestion. You are correct, qRT-PCR can be considered complementary to proteomic data. In this study, qRT-PCR is also utilized to complement proteomic data. Our qRT-PCR analysis is based on previous studies that employed qRT-PCR to validate the reliability of proteomic results (Xin W, Zhang L, Gao J, Zhang W, Yi J, Zhen X, Bi C, He D, Liu S, Zhao X. Adaptation mechanism of roots to low and high nitrogen revealed by proteomic analysis. Rice. 2021, 14: 1-4; Zhang D, Liu J, Zhang Y, Wang H, Wei S, Zhang X, Zhang D, Ma H, Ding Q, Ma L. Morphophysiological, proteomic and metabolomic analyses reveal cadmium tolerance mechanism in common wheat (Triticum aestivum L.). Journal of Hazardous Materials. 2023, 445:130499). Additionally, there have been studies conducted on various other plants using qRT-PCR to validate the reliability of proteomic results. To verify the reliability of the proteome data, we will randomly select differentially abundant proteins for analysis through western blot or identification using LC-MS/MS in follow-up experiments.

  1. In Figure 1A, an appropriate scale bar has to be added to the photos of both WT and mtn1.

Response: Thank you for your valuable comment. The scale bar has been added to Figure A. Changes were highlighted in the revised version in line 115.

  1. Use either the term „tillering stage“ or „tiller development“ but not „tillering development“ including the manuscript title: „Physiological and proteomic analysis of a mtn1 (more tillering number 1) mutant reveal key players in centipedegrass tiller development“

Response: We replaced the term "tillering development" with "tiller development" throughout the article. Changes were highlighted in the revised version.

  1. Use the term „DAPs“, i.e., „differentially abundant proteins“ instead of „DEPs“, i.e., „differentially expressed proteins“ since the differences in protein relative abundance always represent a result between protein biosynthesis („protein expression“) and protein degradation.

Response: Thank you very much for your valuable suggestion. In this article, the term "differentially expressed proteins" (DEPs) has been replaced with "differentially abundant proteins" (DAPs). Changes were highlighted in the revised version.

  1. Line 321: Correct the term „phycological data“ to „physiological data“ since „phycological“ means associated with algae.

Response: Thanks for this good suggestion. We substituted the term "phycological data" with "physiological data". Change was highlighted in the revised version in line 544.

  1. In Conclusion, a new figure (Figure 10) providing a model scheme comparing the molecular mechanisms involved in tiller development between centipedegrass WT and mtn1 mutant based on physiological and proteomic results of the present study.

Response: Thanks for your comment. We have included this figure and its corresponding legend in the conclusion section. Changes were highlighted in the revised version in line 571.

  1. Introduction, line 53: Add a comma following the word „Recently,….“

Response: Thanks for this good suggestion. We added a comma following the “Recently”. Change was highlighted in the revised version in line 80.

  1. Results, Line 78: Modify the word form „In consistent“ to „In consistence“ in the statement: „In consistence with the chlorophyll contents,…“

Response: Thanks for your comment. In order to make the sentence more concise, we exchanged this sentence as follows: The net photosynthetic rate of the mtn1 mutant increased by 31.44% compared to the WT, consistent with the chlorophyll contents (Figure 2C). Change was highlighted in the revised version in line 121.

  1. Discussion, line 215: Modify the verb form „served“ to the present form „serves“ in the statement: „The BIG protein serves as an auxin efflux carrier…“

Response: Thanks for your comment. In the revised version, we have modified the sentence as follows: "while BIG and AUX1/LAX proteins serve as auxin efflux carriers and influx carriers, respectively [30-33]." Changes were highlighted in the revised version in line 320.

  1. Line 220: Use the word form „enhanced auxin transport“  instead of „more auxin transport.“

Response: Thanks for your good comment. To enhance clarity of the discussion, we revised this sentence to “The simultaneous high protein abundance of these auxin synthesis and transport proteins led to lower auxin levels in the mtn1 mutant [14].” Changes were highlighted in the revised version in 320-322.

  1. Line 226: Correct the spelling in „GAs“ not „Gas“ for gibberellins.

Response: Thanks for your good comment. The word in the manuscript was modified.

  1. Line 251: Modify the word form accumulated“ to „accumulation“ in the statement: „The increase in the accumulation of these proteins may cause an increase in soluble sugar content in tiller buds of mtn1 mutant.“

Response: Thanks for your comment. We modified this sentence as you suggested. Change was highlighted in the revised version in line 357-358.

  1. Line 321: Correct the term „phycological data“ to „physiological data“ since „phycological“ means associated with algae.

 Response: Thanks for this good suggestion. The “phycological data” was substituted with “physiological data”. Change was highlighted in the revised version in line 544.

Sincerely,

Ling Li

Jiangsu Province and Chinese Academy of Sciences

Zhongshanmenwai Qianhuahoucun, No. 1, Nanjing 210014, China

Round 2

Reviewer 1 Report

Comments and Suggestions for Authors

After a second reading, I believe that most of my earlier comments have been implemented. However, I feel that there are important points to improve regarding the background to the introduction and the discussion. Also in the figure format. Statistics that were omitted from the first version. And some mistakes. Otherwise, I feel that the manuscript has been improved and that it can be published after these changes. Please review the comments in the attached file.

Comments on the Quality of English Language

In my opinion, English is fine.

Author Response

Responses to Reviewer Dear reviewer: Thank you for your valuable comments on this manuscript. The manuscript has been thoroughly revised based on the reviewer's comments and suggestions, addressing each point in detail. 1. Line 29 I think it should clearly indicate the existing information on species close to centipedegrass, and then expand it to other species. If there is no information on nearby species, indicate so. Response: We appreciate your valuable comment and suggestion. Following your suggestion, we have included information on species closely related to centipedegrass in the Introduction section. Changes were highlighted in the revised version in line 47–64. 2. Line 35 Reference? Response: Thanks for this good comments. We have included the appropriate reference and highlighted changes in the revised version in line 34–35. 3. Line 46 Some bibliography with species related close to centipedegrass? Response: Thank you for your valuable comment. Based on your suggestion, we have included information on species closely related to centipedegrass, including centipedegrass, switchgras. Changes were highlighted in the revised version in line 47–51. 4. Line 48 mtn1 mutant in centipedegrass Response: Thanks for this good comments. We added “in centipedegrass” after mtn1 mutant. Changes were highlighted in the revised version in line 53. 5. Line 51 Can you include them here? Response: Thank you for your valuable suggestion. We have provided the detailed species names and revised the sentence as follows: “(GAs) play a role in regulating tillering/branch development, but their effects vary across species, such as centipedegrass, tall fescue, poplar and sweet cherry [11, 14–16].” Changes were highlighted in the revised version in line 56. 6. Line 52 Same previous comment. I think it is important to know first the knowledge in close species to centipedegrass. Extensible to the rest of the hormones. Response: Thanks for this good comments. Based on the reviewed references, it has been found that GA can stimulate branching in woody plants. To date, no reports have been found indicating that GA can promote branching development in species similar to centipedegrass and other grasses. 7. Line 53 I think this should be introduced first when you talk about GAs Response: Thank you for your good suggestion. This information is placed at the forefront. Changes were highlighted in the revised version in line 57–64. 8. Line 54 Reference? Response: Thanks for this good comments. We included the reference in our study, and changes were highlighted in the revised version in line 64. 9. Line 60 From here I think I should start a new paragraph by stopping to talk about hormones and starting to talk about sugar. Response: Thanks for this valuable suggestion. As you suggested, we have started a new paragraph here. Changes were highlighted in the revised version in line 74. 10. Line 62 Maybe this sentence should be placed first in the new paragraph. Response: Thanks for this good comments. We have placed this sentence as the first in the new paragraph. Changes were highlighted in the revised version in line 74. 11. Line 66 References? Response: Thanks for this good comments. We included the reference in our study, and changes were highlighted in the revised version in line 82. 12. Line 70 Was this publicated in a previous research ? Response: Thanks for this comment. We possess a patent for the cold plasma mutagenesis technology mentioned, and we have conducted a preliminary study on this subject. We have included the published paper in the manuscript. Changes were highlighted in the revised version in line 87. 13. Line 90 The letters that correspond to each figure (top-left) should follow the format of the template. Response: Thanks for your good suggestion. We have revised Figure 1 according to the template format. Changes were highlighted in the revised version in line 104. 14. Line 103 The size of the letters in the figures is not uniform. Response: Thank you for your suggestion. We redrawn the figure, and keep the letters size consistent. Changes were highlighted in the revised version in line 119. 15. Line 116 etc.? Response: Thanks for this comment. We removed the word. Change was highlighted in the revised version in line 134. 16. Line 121 Please also include abbreviations and try to improve the quality of the figures by also increasing their size. The format of the text and numbers must be uniform 'Palatino Linotype'. The signs (-), an orthographic sign, should not be confused with the minus sign, which is why this must be changed throughout the manuscript to an dash (–), also in the figures. Response: Thank you for your good suggestions. As you suggested, we have made several modifications: redrawn the figures, added abbreviations, changed the font to 'Palatino Linotype', corrected the signs (–). Changes were highlighted in the revised version. 17. Line 126 Add the method used. Response: Thank for this comment. The method of PCA analysis was explained in the Materials and Methods section, specifically in lines 504–506. 18. Line 144 Please, try to include quantitative differences. Please, revise carefully the manuscript. Response: Thank you for your valuable suggestion. We have added the quantitative differences and revised the manuscript. Changes were highlighted in the revised version in line 162–180. 19. Line 146 significant enrichment Response: Thank for this comment. We have revised the manuscript. Changes were highlighted in the revised version in line 172. 20. Line 148 Response: Thank for this comment. We have revised the manuscript. Changes were highlighted in the revised version in line 174. 21. Line 149 Response: Thank for this comment. We have revised the manuscript. Changes were highlighted in the revised version in line 177. 22. Line 153 Why is the information related to statistics suppressed? Response: Thanks for this comment. Considering that this figure represents an enrichment analysis diagram, we have made the decision to reintroduce the significance mark and include a description of the analysis method in the materials and methods section. Changes were highlighted in the revised version in line 182–187, line 486–490. 23 Line 154 Why is the information related to statistics suppressed? Response: Thanks for this comment. Considering that this figure represents an enrichment analysis diagram, we have made the decision to reintroduce the significance mark and include a description of the analysis method in the materials and methods section. Changes were highlighted in the revised version in line 202–204, line 486–490. 24. Line 174 Check the table as it indicates Table 2, it also appears that some figure is missing, and some of the text is underlined in yellow. Also try to improve the legend if possible. Response: Thanks for your good suggestions. We meticulously reviewed all tables, making changes to the table names, order, and typeface, and eliminating the yellow shading. In Figures 7 and 8, we identified the differentially abundant proteins that exhibited trends consistent with the phenotype. 25. Line 194 Italics Response: Thanks for this comment. We modified this word into the italics. Change was highlighted in the revised version in line 230. 26. Line 197 In Table S2 it indicates Table S3, please revise it. Response: Thanks for your good suggestion. We revised all the tables. 27. Line 209 Change to S3. In the table it indicates S1 instead of S3. Response: Thanks for your good suggestion. We revised all the tables. 28. Line 213 Details of statistics is not included. Response: Thanks for this comment. We added the statistics as follows: “Data are shown as mean ± standard error. Different lowercases indicate significant differences determined by Duncan’ tests (p < 0.05).” Changes were highlighted in the revised version in line 272. 28. Line 218 I think it should be enriched with more background indicating similarities or differences related to the obtained results and if possible in related species. Response: Thanks for your good suggestions. We have revised the discussion section as you suggested and compared the results of our study with those of previous studies. Changes were highlighted in the revised version in line 316–363. 29. Line 225 Can you explain the connexion of this sentense with the previous text? Response: In our previous article, we analyzed the levels of gibberellin (GA) and auxin in the tillers of mtn1 mutant and wild-type in centipedegrass, and observed a decrease in their concentrations. The mechanism underlying their reduction remains unclear. This study identified differentially abundant proteins associated with auxin, GAs and BRs, which elucidate the underlying mechanism of their content alterations. This study builds upon prior research conducted in a related study. 30. Line 227 Please, revise my previous comments. Response: Thank you for your good comments. We revised this sentence as follows: “Plant hormones, including auxin, GAs and BRs, along with sugars play a crucial role in regulating tiller development in plants. Therefore, we propose that these factors are significant in regulating tiller development in suggesting the potential regulatory roles of plant hormones and sugars in centipedegrass tiller development [8, 26].” Changes were highlighted in the revised version in line 305–309. 31. Line 289 Do you know the texture? Response: The texture of the soil is clay loam. 32. Line 304 Please, include units of measurement. Extensible to other points. Response: Thank you for your suggestion. We have revised this section. Changes were highlighted in the revised version in line 409–411, 423–436. 33. Line 351 In brackets. Please, include city if possible. Response: Thank you for your good suggestion. We have added the city in the brackets in line 452. 34. Please, include the information of the PCA. Also the information related to those supprimed in the figures. Also information related to the normality and homocedasticity of data. Response: Thanks for this comment. We added the PCA analysis method and statistical analysis method in this section as follows: Principal component analysis (PCA) was conducted on the identified proteins using the ggplot2 package in R language software (v 3.5.0, R Core Team, Copenhagen Busi-ness School, Frederiksberg, Denmark). The ANOVA statistical analysis was performed using SPSS 16.0 software (International Business Machines Corporation, Armonk, USA). Data visualization was carried out using Excel 2021 and PPT 2021 (Microsoft Corporation, Redmond, USA). The student-Newman-Keuls test and Tukey’s multiple comparison test were performed using SPSS 16.0 software (International Business Machines Corporation, Armonk, USA) to analyze variations in the mean of two samples and multiple samples, respectively. Changes were highlighted in the revised version in line 504–512. 35. Line 419 I think this should be integrated in Results. I should be also improved the legend. Response: Thank you for your good comments. The relevant information has been included in the Results section and the legend has been enhanced. Additionally, the conclusion section has been revised. Changes were highlighted in the revised version in line 274–293, 527–529.

Reviewer 2 Report

Comments and Suggestions for Authors

Dear Authors,

I have no further comments on the revised manuscript. I can recommend the manuscript publication in Plants.

Final recommendation: Accept.

Author Response

Responses to Reviewer

Dear reviewer:

Thank you very much for your attention on our manuscript (plants-2903077) entitled “Physiological and proteomic analysis of a mtn1 (more tillering number 1) mutant reveal key players in centipedegrass tillering development”. We thank you very much for the comments on our manuscript.